# One-Step Preparation of Both Micron and Nanoparticles

**DOI:** 10.3390/polym16223120

**Published:** 2024-11-07

**Authors:** Zihao Guo, Zhiyuan Zhang, Yunchen Cao, Chunyi Chen, Juan Wang, Haoran Yang, Wenbin Song, Yiyang Peng, Xiaowei Hu

**Affiliations:** 1School of Chemistry & Chemical Engineering, Linyi University, Linyi 276000, China; 13562196795@163.com (Z.G.); 19553526658@163.com (Z.Z.); change10086@outlook.com (C.C.); 13020540405@163.com (H.Y.); 19953709163@163.com (W.S.); 17771553578@163.com (Y.P.); 2Linyi Hongrun Environmental Testing Co., Ltd., Linyi 276000, China; hanyuefuqu@163.com

**Keywords:** micro/nano dual-size particles, soap-free emulsion polymerization, one-pot medium, coating

## Abstract

The complex materials comprised of both micron and nanometer-sized particles (MNPs) present special properties different from conventional single-size particles due to their special size effect. In this study, the MNPs could be simultaneously synthesized in a one-pot medium by soap-free emulsion polymerization, without harsh preparation conditions and material waste. In the whole process, the amphipathic siloxane oligomers would migrate to the mixed monomer droplet surface to reduce the surface energy of the system and further complete hydrolysis–condensation to obtain the SiO_2_ shell at the water–oil interface. On the one hand, the mixed monomers inside the above shell would migrate outward driven by the capillary force generated at the shell mesopore and be further initiated by the water-soluble initiator potassium persulfate (KPS), resulting in the formation of bowl-shaped micron particles with “lunar surface” structure. On the other hand, the residual mixed monomers dissolve in water and could be polymerized by initiating free radicals in the water phase to obtain popcorn-like nano-sized particles. The above two particles are clearly displayed in the SEM photos, and the DLS characterization further shows that the sizes of two particles are concentrated at 1.4 μm and 130 nm, respectively. Interestingly, the uniformity of obtained particles has a great relationship with the added amount of BA, and the perfect MNPs would appear when the St/BA feed mass ratio is 1:2. Moreover, the MNPs exhibit film-forming property, and the SiO_2_ component is evenly distributed in the formed coating. Thus, this study is not only beneficial to the theoretical research of soap-free emulsion polymerization but also to the application of multifunctional coatings.

## 1. Introduction

Generally speaking, micron and nano-sized particles have been widely studied and developed in many fields by virtue of their respective structural and performance advantages [1,2,3,4,5]. In recent years, a large number of researchers have found that the materials formed by micro/nano dual-size particles (MNPs) present special properties different from conventional single-size particles in terms of sound, light, electricity, magnetic, and thermal; therefore, they show outstanding application potential in the fields of superhydrophobic, electromagnetic wave absorption, biomedicine, and multifunctional coating [6,7,8,9,10,11]. The MNPs could be mainly divided into two types: one is a simple blend of the micron and nanoparticles; the other is that the MNPs co-exist in a multistage structural particle.

In view of the excellent properties and application advantages of MNPs, multitudes of preparation methods have been developed in recent years. Among them, the physical blending of two-dimensional particles is the most common method [12,13]. The main steps are to first prepare the micron and nano-sized particles, respectively, and then blend them in proportion to prepare complex materials and explore their application properties. However, this method is limited in practical industrial production because of its complicated steps and unequal distribution of the two-size particles. Based on the above problems, many methods have been developed to prepare hierarchical structural particles with uniform distribution of micron and nano dual-size materials by physical or chemical etching [14,15,16]. For example, the Ma research group [15] has successfully prepared a multistage structural aluminum surface with superhydrophobic and radiation-resistant properties, which were realized by etching it with laser. Although the above method could prepare a uniform micro-nano structure, it would cause great material waste in the preparation process and be difficult to prepare MNPs of organic polymer components. In the emulsion polymerization system, the in-situ preparation of uniform MNPs by surface nucleation came into being [17,18,19,20]. The main steps are as follows: first, micrometer-size particles are prepared in the emulsion system, and then these particles are used as the core to further initiate the polymerization of the second monomer to form nanoscale particles on the micrometer-size particle surface, resulting in the appearance of a micro-nano hierarchical structure. The existence of emulsifiers that can reduce the surface energy of the system is beneficial to the preparation of particles in the early preparation stage, but they are difficult to remove completely, and residual emulsifiers could further degrade the properties of the material in the later application process. Later, the development of soap-free emulsion polymerization is expected to solve the above problems [21,22,23,24]. For example, Pan et al. [21] used the soap-free seed emulsion polymerization method, using non-crosslinked P(VC-co-AAEM) particles as seeds and hydrophilic groups as stabilizers, to prepare uniform “strawberry” particles with micro-nano hierarchical structure. In the existing soap-free emulsion polymerization methods, the research of MNPs focuses on the realization of micro-nano structure on one particle and rarely produces two particles with strict separation of micron and nano size in a pot of media at the same time, let alone the application of such mixed particles.

Herein, it was accidentally discovered that a novel and facile soap-free emulsion polymerization, without harsh preparation conditions and material waste, could be used to simultaneously synthesize micron particles (about 1.4 μm) and nanoparticles (about 130 nm). The method for preparing MNPs was based on the in-depth study of this phenomenon and offered several distinct advantages, including the following: (i) This method provided a new idea for the preparation of micro/nano dual-size materials. In our entire process, neither cumbersome steps nor special equipment would facilitate the large-scale fabrication of MNPs. As far as we know, almost no one has reported a similar way. (ii) No additional emulsifiers were added throughout the process, and the oligomers from in-situ prehydrolytic polycondensation of silane precursor MTES could be used to stabilize the whole soap-free emulsion system, which avoided the negative impact of small-molecule emulsifiers on the final particle properties; (iii) The MNPs exhibited good film-forming properties, and the SiO_2_ component could be evenly distributed in the formed poly(styrene-co-butyl acrylate) (SA) coating. Thus, the discovery of this special phenomenon had not only enriched the preparation methods of micro/nano dual-size materials but also had the application potential in the multifunctional SA coating.

## 2. Experimental Section

### 2.1. Materials and Treatment

The styrene (St, 104.15, above 99.5%) and butyl acrylate (BA, 128.17, above 99.5%) were purchased from Tianjin Chemical Reagent Co., Ltd. in Tianjin, China. In order to remove impurities from the above monomers, a vacuum distillation at 70 °C was required before use. Potassium persulfate (KPS, 270.32, 99.5%, China Medicine Group Chemical Reagent Co., Ltd. in Shanghai, China) was used with no further purification. Methyltriethoxylsilane (MTES, 178.30, 98%) was purchased from Shanghai Macklin Biochemical Co., Ltd. in Shanghai, China. Hydrochloric acid (HCl, 36.46, 37 wt%) and ammonium hydroxide aqueous solution (NH_3_∙H_2_O, 36.46, 28 wt%) used were analytical grade.

### 2.2. Polymerization Procedure

The preparation of latex particles: The whole preparation process of composite particles mainly included the hydrolysis condensation process of MTES and the copolymerization stage of mixed monomers. The typical experimental details are as follows: the whole reaction site was in a 100 mL four-necked flask with a mechanical stirrer, a nitrogen gas inlet, a condenser, and a thermometer. First, the monomer mixture comprised 1.4 g MTES, 0.4 g St, and 0.8 g BA, which were ultrasonically dispersed at 40 Hz for 5 min and then dripped into 50 mL of acidic aqueous solution with a pH value of 3.5 at an addition rate of 0.4 g/min. The above mixture was stirred at a rate of 250 rpm for 1 h under room temperature, then adding a certain amount of ammonia water to adjust the pH value to about 10. The SiO_2_ shell with a mesoporous structure containing St and BA monomers inside the shell was formed after the MTES continued the hydrolytic condensation for 50 min. Then, the reaction system was heated from room temperature to 70 °C in the nitrogen atmosphere. Both micro- and nano-sized particles were prepared simultaneously after a further reaction of 2 h induced by the water-soluble initiator KPS. In addition, the sample aliquots were taken every hour throughout the reaction process to observe particle morphology.

The preparation of particulate film: Both micron and nanoparticles were centrifugally washed twice with ethanol/water (2:1 by volume) to remove the remaining monomers, initiator, etc., finally re-dispersed in deionized water. Then, the above dispersion was dropped onto a clean glass plate, and the particulate film was obtained successfully by vacuum drying for 8 h. The morphology of the obtained particle films was observed under an optical microscope.

### 2.3. Characterization

Morphological characteristics of composite particles were studied by a field-emission scanning electron microscope (FE-SEM, Nano 450 operated at 10 kV, FEI, Hillsboro, OR, USA) and a transmission electron microscope (TEM, H-7650B at 80 kV, Hitachi, Tokyo, Japan). As for the SEM sample, a few drops of latex were diluted with deionized water to gain a translucent suspension, which was ultrasonicated for 40 min. Then, a drop of the suspension was cast onto a conductive silicon wafer and dried in the freeze dryer for 24 h. For preparation of TEM samples, the same steps were used except replacing the conductive silicon wafer with a 400-mesh copper grid coated with carbon layer. The sizes and size distributions of the latex particles in aqueous medium were measured by a Zeta-Sizer 90 dynamic laser scattering particle size analyzer (DLS, Malvern, UK). The wavelength of incident light was 532 nm, and the scattering angle was 90°.

## 3. Results and Discussion

The P(St-BA)/SiO_2_ composite particles with both micron and nanometer size could be simultaneously prepared in a one-pot medium by soap-free emulsion polymerization, using MTES pre-hydrolyzed polycondensation product as stabilizer, and the detailed preparation process was shown in Figure 1. Furthermore, the formation mechanism, detailed structural information, and film-forming properties of the above particles are characterized and verified below.

It was pleasantly found that micron and nano-sized particles co-existed in a pot of media, and the related morphological results are shown in Figure 1A. To more clearly display the detailed morphological characteristics of the above particles, the obtained particles were placed in a centrifuge tube and washed them by centrifugation with ethanol. Herein, the micron-sized composite particles were deposited at the bottom of the centrifuge tube, while the nano-sized particles were located in the upper suspension. Then, the SEM was used to characterize the apparent morphology of the nano- and micron-sized particles, and the results are shown in Figure 1B and Figure 1C, respectively. Among them, the nano-sized particles in the Figure 1B had a uniform popcorn-like shape, while the micron-sized particles in the Figure 1C showed a uniform bowl-shaped morphology with an uneven structure similar to the “lunar surface”. To further confirm that the above composite particles of micron and nano size could be prepared simultaneously in a one-pot medium, the size and size distribution of the obtained particles were characterized by DLS, and the results are shown in Appendix A. First, the particle size distribution showed a strict bimodal state, which further verified the coexistence of micro- and nano-sized particles. Then, the particle size of the nano-dimensional particles in the system was concentrated around 130 nm, and the average particle size of the microsized particles was about 1.4 μm, which proved that both particles were relatively uniform.

The EDS test was used to further determine the elemental composition of both particles, and the corresponding result was shown in Figure 2. A spot on a micron particle was selected for elemental analysis (Figure 2B); the result suggested a uniform distribution of C, O, and Si elements on this particle, and the element percentages, respectively, were 22.48 wt%, 18.29 wt%, and 59.23 wt%. The nano-sized particles were too small to be selected at a certain point; therefore, we have selected a region to analyze their element data. And Figure 2C displayed that there were C (30.04 wt%), O (28.71 wt%), and Si (41.25 wt%) elements within nano-sized particles. The above results showed that both micron and nano-sized particles were P(St-co-BA)/SiO_2_ composites. In addition, the difference in element percentage between both particles might be due to the following reasons: (i) The formation process of micron and nano-sized particles with different morphology characteristics might be different, which could lead to certain differences in the content of organic P(St-co-BA) and inorganic SiO_2_ components in the two sized particles; (ii) There were differences in the positions of the two particles selected for elemental content analysis, that is, a spot on micron particles while one region on nanoparticles.

The key steps of successful preparation of MNPs mainly included hydrolysis, polycondensation of MTES, and polymerization procedures of St and BA monomers. To further explore the particle formation mechanism, the particles formed in the above two stages were analyzed and discussed, respectively. Firstly, based on my previous work, we have drawn a conclusion that the siloxane oligomers could show amphiphilic properties like emulsifiers to emulsify the pure St monomer phase [25,26]. Here, in order to further determine whether the MTES prepolymers also have stable emulsification on St and BA-mixed oil phase monomers, we have taken the addition of monomer as a variable to study the morphology characteristics of hydrolyzed polycondensation products of MTES. The obtained results are shown below. When no reactive monomer phase was added to the system, the particles obtained by hydrolytic polycondensation of MTES in water were uniform spherical shapes (Figure 3(iA)), and the TEM image in Figure 3(iiA) showed that the interior of the particles was solid. However, when St and BA mixed monomers were added to the system, the bowl-shaped particles were obtained after hydrolysis and condensation of MTES in the system (as shown in Figure 3B). When the above mixed monomers were replaced by pure St, spherical particles with hollow structures were successfully prepared, as shown in SEM and TEM images in Figure 3C. Based on the analysis of the above results, the following conclusions could be drawn: On the one hand, oil-soluble monomers added to the water phase could increase the surface energy of the system, resulting in partial amphiphilic siloxane oligomers formed during the hydrolytic polycondensation of MTES that would migrate to the O/W interface to stabilize the whole system. After further reaction of the above oligomers located at the monomer droplet surface, the silicon-based shell with monomers inside could be obtained. On the other hand, the difference in water solubility of different monomers could result in a change in the monomer droplet size, thus determining the thickness of the above SiO_2_ shell. As we all know, the water solubility of St was worse than that of BA, which could lead to the surface energy of the system increasing with the addition of St monomer. When pure St was added to the system, more amphiphilic siloxane oligomers would be distributed on the surface of the monomer droplet to reduce the surface energy, so that the SiO_2_ shell thickness was large in this case, and a uniform hollow structure could be obtained once the internal monomer was removed, as shown in Figure 3C. When the added monomer phase contained BA monomer, the SiO_2_ shell size would become larger and the shell thickness was smaller. Furthermore, once the monomer phase inside the shell was removed, the shell would easily collapse and form a concave structure, as shown in Figure 3B.

Figure 3 shows the studied morphology characteristics of particles obtained by removing the monomer phase inside the SiO_2_ shell. We were curious about what would happen if the monomer phase inside the shell was further induced polymerization by the water-soluble initiator KPS. The relevant experiments were further carried out, and the results are shown in Figure 4: When the monomer phase in the system was pure St, the obtained particles had a relatively uniform strawberry structure (Figure 4A) after triggering the monomer reaction inside the SiO_2_ shell. When the total amount of monomer phase added in the system was unchanged and the ratio of St to BA was 1:2, both micro particles with a bowl-shaped “lunar surface” and nanoparticles with a popcorn-like structure simultaneously appeared in the system (Figure 4B). Thus, the presence of BA had a great influence on the morphology and size of the obtained particles. Furthermore, the experiments with different feed ratios of St and BA were further designed to study the effect of BA content on the structural characteristics of the final particles. When the mass ratio of St to BA was 1:1, the obtained particles had a rough hemispherical structure (Figure 4B). When the addition amount of St and BA was 1:3, the final particles in Figure 4D were still the coexistence of micron and nanometer, but the uniformity of two types of particles became worse. When the added monomer in the system was pure BA, the particles obtained after BA polymerization were very heterogeneous and viscous. Through the analysis of the above results, the following conclusions could be drawn: The addition of BA monomer in the soap-free emulsion polymerization system of St was the key to the successful preparation of micron and nanometer two-size particles. Moreover, the addition amount of BA determined the uniformity of the two sizes of particles, and stable MNPs could be obtained when the St/BA feed mass ratio was 1:2.

Based on all the above experimental results and discussions, we proposed the possible formation mechanism of MNPs (Figure 1) formed in a one-pot medium; the two key steps are shown in Figure 5. (i) The hydrolysis polycondensation process of MTES: The siloxane oligomers (Figure 5A) obtained by prehydrolyzed polycondensation of MTES showed amphiphilic property due to the existence of hydrophobic methyl, ethoxysilane, and hydrophilic silanol groups [25,26]. For the above oligomers, most would migrate to the O/W interface to reduce the surface energy of the system (as shown in Figure 5B), and some remained in the water phase. After the system was adjusted to the basic condition, the oligomers at the water–oil interface could further complete hydrolytic polycondensation to form an SiO_2_ shell containing St and BA monomers inside (Figure 5C), and this phenomenon has been confirmed in Figure 3. For the mixed monomer phase, in addition to the existing in the above shell, the rest is located in the water phase due to the certain water solubility of BA monomer. (ii) The polymerization process of St and BA mixed monomer phase: The water-soluble initiator KPS was added to the above system to trigger the copolymerization of St and BA. For the monomers inside the shell: As shown in Figure 5D, on the one hand, the mesoporous channels on the SiO_2_ shell could generate a strong capillary force to induce the internal monomers to migrate outward and polymerize [27]; On the other hand, the osmotic pressure could be generated when the St and BA monomers were polymerized through KPS initiation, which would provide an additional impetus for continuous transport of the monomer. Furthermore, the collapse could occur on the SiO_2_ shell once the St and BA monomers migrated from the inside out, which was the conclusion drawn by the information in Figure 3B. In addition, the surface energy of the system would increase as the copolymer chain grew, resulting in the polymer components not completely covering the SiO_2_ shell to expose some hydrophilic silicon hydroxyl group. Thus, micro particles with bowl-shaped “lunar surface” morphology in above Figure 1C were successfully obtained in the system. For the monomers dispersed in the water phase, they could also be polymerized by water-soluble initiator KPS. As shown in Figure 5E, when the polymerization degree of monomers was low, the hydrophilic groups ([–OSO_3_]^−^) brought by KPS could stabilize the particles [28]. As the size of latex particles increases, the siloxane oligomers could also migrate to the latex particle surface to reduce the interface tension, keeping the system stable. There were few dispersed monomer molecules and many reactive sites initiated by active free radicals in the aqueous system, which resulted in the formation of smaller particles in this state. Thus, nanoparticles with a popcorn-like structure were successfully prepared, as shown in the above Figure 1B. As a result, both micron and nano-sized composite particles appeared simultaneously in the soap-free emulsion polymerization system; however, the above two kinds of particles would have certain differences in element composition due to the difference of their reaction sites and processes, which was consistent with the information shown in Figure 2.

As is known to all, the styrene acrylic (SA) emulsion is an important part of a common coating, and SiO_2_ is often used as a coating additive to increase its wear resistance, corrosion resistance, etching resistance, or other properties [29,30]. However, in the commonly used SiO_2_ modified coating methods, SiO_2_ particles are easy to agglomerate, which could make the final coating performance worse [31,32,33]. In order to explore the potential application value of the obtained dual-size co-existing P(St-BA)/SiO_2_ composite particles in the field of coating, these particles were used to prepare coating at 50 °C and further observe it under an optical microscope. As shown in Appendix A, the coating surface was relatively flat, which could be inferred that the MNPs had excellent film-forming performance. In addition, after the P(St-BA) component formed a membrane, the residual micron-level SiO_2_ component could be uniformly dispersed in the formed film. Therefore, the as-obtained MNPs were expected to show potential application value in the field of functional coatings by virtue of their excellent film-forming properties and the outstanding performance advantage combined organic and inorganic components.

## 4. Conclusions

In summary, we have simultaneously fabricated MNPs in one-pot medium based on a new soap-free emulsion polymerization method. The above two particles have uniform sizes of 1.4 μm and 130 nm, respectively. The important reason for the system’s stability is that the siloxane oligomers obtained by prehydrolyzed polycondensation of MTES showed amphiphilic property due to the existence of hydrophobic methyl, ethoxysilane, and hydrophilic silanol groups. The main reason for the simultaneous appearance of micron and nanometer particles is that the polymerization of the mixed monomer phase occurs in different places in the system. On the one hand, some monomers dissolved in water were polymerized by KPS to obtain nanoscale composite particles. On the other hand, the oil-soluble mixed monomers located inside the SiO_2_ shell formed by the above siloxane oligomers continuously migrate to the shell outside and react to form micron composite particles. Furthermore, the MNPs have the potential to be used in functional coatings due to their good film-forming property and their ability to solve the problem of aggregation of siloxane inorganic components in organic coatings.

## Data Availability

Data are contained within the article and Appendix A.

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
