# Peer review of "One-Step Preparation of Both Micron and Nanoparticles"

_polymers, 2024, doi:10.3390/polym16223120_

Round 1

Reviewer 1 Report

Comments and Suggestions for Authors

In this manuscript, the author studied the one-step preparation of both micron and nanoparticles. The manuscript should be accepted after addressing the following issues;

1). In the abstract, the authors should improve this part by adding more information related to the findings and adding some numerical values to improve the novelty.

2). In the introduction, the authors described the text and general information related to the topic. The authors add more relevant information and findings to improve the worth of the manuscript.

3). In the material and methods section; the authors should more information related to the purity, source molecular weight, etc.

4). 2.2. Polymerization Procedure; better the authors should add some flow chart or schematic diagram of the synthesis procedure

5). XRD is a key tool to study the nanocomposite structure and other information such as crystalline size etc. The authors should measure the XRD

6). The authors mentioned the JCPDS card number and Rietveld refinement for the successful synthesis of the materials

7). In Figure 1; the authors should improve the caption of the Figure. Use the same number of capital or small

8). In Figure 2, why the percentage ratios are different in both samples; the authors should explain the reason

9). There are some grammatical and typos errors in the manuscript. The authors check it carefully and improve it.  

Comments on the Quality of English Language

There are some grammatical and typos errors in the manuscript. The authors check it carefully and improve it.  

Reviewer 2 Report

Comments and Suggestions for Authors

The study presents an investigation on the fabrication of complex materials comprised of both micron and nanometer size particles (MNPs), produced in one-pot medium based on a soap-free emulsion polymerization method. The MS is of high interest to the readers of MDPI Polymers, and may be useful in view of both further research on soap-free emulsion polymerization, and of possible application prospects, e.g. production of multifunctional coatings.

Before accepting the manuscript however, some minor issues have to be addressed:

1. The quality of Figs 2a-c is insufficient; the text is too small and difficult to read.

2. The physicochemical background of the observed phenomena (e.g. interaction mechanisms, time evolution, etc.) should be clarified in more details in view of structure-property relationships of the proposed specific formulations (pp.6-7).

Round 2

Reviewer 1 Report

Comments and Suggestions for Authors

Accepted in the present form

Comments on the Quality of English Language

okay